# Rapid Sorting of Fucoxanthin-Producing *Phaeodactylum tricornutum* Mutants by Flow Cytometry

**DOI:** 10.3390/md19040228

**Published:** 2021-04-17

**Authors:** Yong Fan, Xiao-Ting Ding, Li-Juan Wang, Er-Ying Jiang, Phung Nghi Van, Fu-Li Li

**Affiliations:** 1Key Laboratory of Biofuels, Shandong Provincial Key Laboratory of Synthetic Biology, Qingdao Institute of Bioenergy and Bioprocess Technology, Chinese Academy of Sciences, Qingdao 266101, China; fanyong@qibebt.ac.cn (Y.F.); dingxt@qibebt.ac.cn (X.-T.D.); w873644513@163.com (L.-J.W.); jiangey@qibebt.ac.cn (E.-Y.J.); van@qibebt.ac.cn (P.N.V.); 2University of Chinese Academy of Sciences, Beijing 100049, China; 3Shandong Energy Institute, Qingdao 266101, China; 4Qingdao New Energy Shandong Laboratory, Qingdao 266101, China

**Keywords:** flow cytometry, fucoxanthin, heavy ion irradiation, *Phaeodactylum tricornutum*

## Abstract

Fucoxanthin, which is widely found in seaweeds and diatoms, has many benefits to human health, such as anti-diabetes, anti-obesity, and anti-inflammatory physiological activities. However, the low content of fucoxanthin in brown algae and diatoms limits the commercialization of this product. In this study, we introduced an excitation light at 488 nm to analyze the emitted fluorescence of *Phaeodactylum tricornutum*, a diatom model organism rich in fucoxanthin. We observed a unique spectrum peak at 710 nm and found a linear correlation between fucoxanthin content and the mean fluorescence intensity. We subsequently used flow cytometry to screen high-fucoxanthin-content mutants created by heavy ion irradiation. After 20 days of cultivation, the fucoxanthin content of sorted cells was 25.5% higher than in the wild type. This method provides an efficient, rapid, and high-throughput approach to screen fucoxanthin-overproducing mutants.

## 1. Introduction

Fucoxanthin is an allenic carotenoid that consists of a polyene backbone; it is present in Chromophyta, including brown seaweeds and diatoms [1,2,3,4]. This pigment has many pharmacological properties, such as antioxidant, anti-obesity, and anti-diabetes activities [5,6,7,8,9]. Despite its utilization prospects, fucoxanthin application is limited due to its low extraction efficiency from seaweeds and difficult synthesis via chemical means [10]. The marine microalgae *Phaeodactylum tricornutum* is a suitable platform to produce fucoxanthin because of its higher fucoxanthin content than brown algae and most other diatoms and its potential to grow rapidly in photobioreactors [11]. There are many studies on the mechanism and pathway of fucoxanthin synthesis in *P. tricornutum*. The available genome information and abundant means of genetic transformation are conducive to genetic engineering and metabolic engineering in the future [12,13,14,15]. Moreover, the structural analysis of the fucoxanthin chlorophyll a/c-binding protein also helps in understanding the significance of fucoxanthin synthesis in promoting photosynthesis [16].

In our previous work, heavy ion irradiation was adopted for microalgae mutagenesis and preparing mutant pools [17]. Heavy ion beams have a high linear energy transfer (LET), which can produce dense ionization along their trajectories and cause complex and irreparable DNA damage [18,19]. Moreover, there may be many changes in metabolic pathways and regulation mechanisms, which has important research value for developing the metabolic engineering of microalgae. We successfully used heavy ion mutagenesis to obtain a library of mutants, and used conventional methods, such as high-performance liquid chromatography (HPLC) and chlorophyll fluorometry, to determine pigment content and investigate the robustness of the mutant strains. However, this approach was time-consuming and not suitable for high-throughput screening [20,21].

Flow cytometry is a widely used technology that is convenient for cell analysis and is highly sensitive [22,23,24]. Previously, sorting methods were established for astaxanthin analysis in yeast (*Xanthophyllomyces dendrorhous*) [25] and microalgae (*Chromochloris zofingiensis*) [26]. In this study, we used flow cytometry to sort *P. tricornutum* mutants with different fucoxanthin contents. We also analyzed the correlation between mean fluorescence intensity (MFI) and fucoxanthin content in the sorted cells. The flow cytometry analysis developed here is an effective, high-throughput approach for estimating fucoxanthin content.

## 2. Results and Discussion

### 2.1. P. tricornutum Has a Specific Emission Wavelength

The fluorescence emission spectra of *P. tricornutum* cells were scanned with a spectrofluorometer with 488 nm excitation, as shown in Figure 1. *Nannochloropsis oceanica* IMET1, *Chlorella* sp., and *Mychonastes afer* were used as negative controls because they do not synthesize fucoxanthin. A peak at 680 nm was detected in these three species. However, in *P. tricornutum*, another peak was found at around 710 nm (Figure 1A). When we performed this test with the pigments that were extracted from these strains, the results revealed that no emission peak at 710 nm was detected (Appendix A). The pigments of *Chlorella* sp. and *M. afer* have two obvious fluorescence emission peaks, which are from chlorophyll *a* and chlorophyll *b*. *P. tricornutum* and *N. oceanica* only have a fluorescence emission peak from chlorophyll *a* due to the extremely low level or absence of chlorophyll *b.* In addition, we carried out TLC for pigment separation; it was visible that the main difference between *P. tricornutum* and other algae strains is the fucoxanthin band (Appendix A). Under blue light (440–485 nm), the bands were observed as red fluorescence of chlorophyll. The pigments of *Chlorella* sp. and *M. afer* showed brighter bands. The fluorescence of chlorophyll *b* may have made the band brighter. No pigment emitted fluorescence at 710 nm.

These results illustrate that the emission fluorescence of *P. tricornutum* cells at 710 nm should not be emitted by a single pigment and is most likely a special fluorescence emission peak of the photosynthetic system complex structure in diatoms. In recent reports, the structures of FCP complex, PSI-FCPI, and PSII-FCPII were analyzed by cryo-electron microscopy [16,27,28,29]. In the separation and purification of these complex structures, fluorescence emission spectra at 77 K were measured using a fluorescence spectrophotometer. Dissolved FCP crystals excite at 436, 465, and 540 nm and the fluorescence emission wavelength is 676 nm [16]. The fluorescence spectrum scan of PSI-FCPI showed that there is a single emission wavelength at about 708 nm, which is similar to our results [27]. The fluorescence emission spectra of PSII-FCPII excite at 425, 459, and 500 nm, measuring at 77 K. The main peak is located at 694 nm [28,29]. In our study, the diatom intact cells were used for scanning at room temperature; two main peaks appeared at 680 and 710 nm. These may come from the emission fluorescence of PSII-FCPII and PSI-FCPI supercomplex, respectively. In particular, the peak at 710 nm is different from fluorescence emission wavelengths in most photosynthetic system, and compared with the pigments in PSII, PSI is more stable in photosynthetic organisms under different culture conditions. Therefore, the fluorescence at 710 nm may more specifically reflect the fucoxanthin content in the cells.

To confirm that this peak corresponded to fucoxanthin content, we designed *P. tricornutum* samples with different fucoxanthin contents. Cellular fucoxanthin production may be blocked using diphenylamine (DPA), which inhibits phytoene conversion to β-carotene, a fucoxanthin precursor [30,31]. Cells were cultured in f/2 medium with different DPA concentrations (0, 25, 50, 75, and 100 μM) for 5 days. In the spectral scans, the MFI differed between the treated samples. As expected, samples with higher fucoxanthin content showed stronger fluorescence intensity at 710 nm. For the samples treated with different concentrations of DPA, the fucoxanthin content correlated well with the fluorescence intensity at the same wavelength (Figure 1B). Therefore, the emission fluorescence at 710 nm was chosen for flow cytometry analysis. In BD FACSAriaⅡ, this wavelength corresponded to the PerCP-Cy5-5 (FL2) channel. Interestingly, although we used the same cell concentration (OD_750_ = 0.4) for the assay, the emission peak at 680 nm also decreased with DPA, indicating that this drug also has an inhibitory effect on chlorophyll synthesis or the structure of PSII-FCPII.

### 2.2. Flow Cytometry Analysis and Sorting

#### 2.2.1. Flow Cytometry Can Be Efficiently Used to Analyze the Content of Fucoxanthin 

We then investigated the relationship between fucoxanthin content (detected by HPLC) and MFI at PerCP-Cy5-5 channel (detected by flow cytometry) in *P. tricornutum* cells treated with different concentrations of DPA (0, 25, 50, 75, and 100 μM) for 5 days. Flow cytometry was used to analyze the samples, and the peak areas in the histogram were integrated and compared. The forward scatter (FSC) signal did not differ among the samples treated with different DPA concentrations (Figure 2A). However, the MFI values of the control group were significantly higher than those of the treated groups (Figure 2B). The fucoxanthin content of these samples was also determined by thin-layer chromatography (TLC) and HPLC. Through TLC detection, we observed that the pigment content of cells changed with the different diphenylamine treatment times. On the fifth day of the treatment, the pigment content of samples treated with different concentrations differed; the color of the cells decreased significantly with rising DPA concentrations (Figure 2C). The determination coefficient of the linear regression was 0.9875 (Figure 2D), indicating a high correlation between fucoxanthin content and MFI.

#### 2.2.2. Flow Cytometry Can Be Used for High-Throughput Separation of Cells with Different Fucoxanthin Contents

Flow cytometry is widely used to evaluate and measure pigment-containing cells during the cell cultivation process in laboratories and industry. Previously, cell analysis and sorting based on astaxanthin content were established for different microorganisms. In the yeast *X. dendrorhous*, the FL4 channel (Beckman Coulter Epics Elite ESP; emission at 665–685 nm) provided the fluorescence intensity that correlated best with the astaxanthin content per cell (correlation coefficient (R^2^) = 0.92) [25]. In the green microalga *C. zofingiensis*, a linear relationship was observed between MFI in FL2 (BD AccuriTM C6 Cytometers, emission at 585 nm) and astaxanthin content [26]. However, the authors of the study did not use this method for cell sorting.

Since flow cytometry can be used to analyze the content of fucoxanthin, can cells with a high content of fucoxanthin still be obtained effectively in the process of sorting? This would play an important role in the screening of industrial strains.

To answer this question, we first blocked fucoxanthin synthesis with DPA in *P. tricornutum* wild type and obtained cells with low fucoxanthin content. We then used the *P. tricornutum* mutant strain as the control group, which was produced through genetic transformation in the previous work. The mutant has the *ble* gene and shows zeocin resistance. The mutant was not inhibited by DPA; thus, its fucoxanthin content was the normal level but higher than in the wild type. Flow cytometry was performed after mixing the two populations. In sorting, we collected the cells with the highest and lowest fluorescence intensities. If the flow cytometry worked well, the highest fluorescence intensity would come from the cells cultured without the DPA inhibitor, which would grow colonies on plates containing zeocin. The cells with the lowest fluorescence intensity would be the wild type cells treated with DPA, which would not grow colonies on plates containing zeocin.

After sorting, approximately 1000 cells were coated onto each plate containing zeocin. In the high fluorescence group, an average of 38 colonies grew per plate (SD = 11.895), while in the low fluorescence group, no colonies appeared (Figure 3). Based on our previous statistics, about 8.0% of the colonies appeared when the cells were directly coated on the plate, and about 5.2% of the cells could grow colonies after flow sorting and coating. These results proved that this method can effectively isolate strains with high fucoxanthin content. The finding that only approximately 3.8% of cells could grow colonies might be due to cell damage and loss during coating and flow sorting. Wild-type cells might also be mixed during sorting. Nevertheless, the sorting success rate was acceptable for our preliminary experiments with high-yield *P. tricornutum* strains.

Next, we used cells irradiated with heavy ions for flow cytometric sorting. Heavy ion irradiation has widely been used to establish a mutant library of plants by microbiologists [32,33]. Mutants with different contents of target products are then screened using a specific index. We sorted the *P. tricornutum* cells irradiated by heavy ions and collected about 8% of the cells with the highest and lowest fluorescence values (Figure 4A,B). In total, 1.5 × 10^6^ cells were collected in each group. Cells containing different contents of fucoxanthin were divided into two groups by flow cytometric sorting and cultured, respectively. After 20 days of cultivation, the fucoxanthin content significantly differed (*p* < 0.05) between the groups (Figure 4C). By cell counting, we found that there was no significant difference in the number of cells in different groups. The number of cells in the positive group (high florescence) was about 6.20 × 10^7^ cells/mL, while that in negative group (low florescence) was about 5.43 × 10^7^ cells/mL. The content of fucoxanthin by unit volume was quite different. It was 37.98 mg/L in the positive group, 29.80 mg/L in the wild-type group and 23.52 mg/L in the negative group. The cells in the positive group contained 6.13 × 10^−7^ μg/cell of fucoxanthin, while those in the negative group had 4.33 × 10^−7^ μg/cell of fucoxanthin (Figure 4D). Compared with the wild type (4.88 × 10^−7^ μg/cell), fucoxanthin content in the positive group increased by 25.5% and the negative group decreased by 11.9%. These results confirm that this method provides a high-throughput approach to screen *P. tricornutum* mutants with high fucoxanthin content. It allows for the rapid evaluation and screening of industry potential mutants in the future through further flow cytometry separation and cell lineage culture.

In our previous report, a simple and accurate method for determining fucoxanthin content in diatom by using spectrophotometry instead of HPLC was established; this method could be conducted in almost all biological and analytical laboratories [34]. However, the spectrophotometry-based fucoxanthin detection method has low flux compared with the method based on flow cytometry. Flow cytometry is a rapid and efficient means to screen mutagenized cells and is a high-throughput approach for rapid analysis and real-time cellular monitoring of specific pigment. These two methods are complementary; furthermore, a rapid high-throughput screening method provides a powerful tool for germplasm screening. This method has good universality for the screening of mutant libraries that rely on synthetic biology platforms, physicochemical mutagenesis, and rational engineering, and is conducive to the selection of high-yield pigment industrial strains.

## 3. Materials and Methods

### 3.1. Strains and Culture Conditions

*P. tricornutum* was stored at the Ocean University of China, Microalgae Culture Center (MACC/B228). *N. oceanica* IMET1 was provided by Feng Chen at the University of Maryland Center for Environmental Science, *M. afer* HSO-3-1 was stored at the China General Microbiological Collection Center (CGMCC 464, Beijing, China) [35,36], and *Chlorella* sp. was screened from natural water by our laboratory and identified by sequencing (the BLAST results after 18S sequencing of algae strain can be found in the Appendix A). *P. tricornutum* and *N. oceanica* IMET1 were cultured in modified f/2 medium, with increased sodium nitrate concentration (1 g·L^−1^). *M. afer* and *Chlorella* sp. were cultured in BG11 medium [17,37,38]. Transformants containing the *ble* gene were transformed with the *p*Pha-T1 vector; they were resistant to zeocin and could grow in the f/2 medium supplemented with 75 μM of zeocin [39].

DPA was used to inhibit phytoene to β-carotene conversion [30,31]. Different fucoxanthin contents in *P. tricornutum* were achieved by supplementing with different DPA concentrations (0, 25, 50, 75 and 100 μM).

*P. tricornutum* cells were irradiated with heavy ions at the Heavy Ion Research Facility at Lanzhou, Institute of Modern Physics, Chinese Academy of Sciences, according to the method previously described [30]. Cells in the early exponential growth phase (ca. 1.5 × 10^7^ cells·mL^−1^) were irradiated with carbon ions, with an energy of 80 MeV·u^−1^, using a 200 Gy ion beam. Afterward, the cells were transferred to f/2 liquid medium and cultured in shaking flasks for 20 days under a photon flux density of 80 μmol photons·m^−2^·s^−1^.

### 3.2. Fluorescence Spectrophotometer and Flow Cytometry Analysis

The fluorescence emission spectra of *P. tricornutum*, *N. oceanica* IMET1, and *M. afer* were scanned with a fluorescence spectrophotometer (HITACHI F-7000, Hitachi, Tokyo, Japan) with a 488 nm exciting light; cellular fluorescence was detected in the 550–750 nm range.

Cytometric analysis and sorting of *P. tricornutum* were performed with a BD FACSAria II (BD Biosciences, San Jose, CA, USA) using a 15 mW argon ion laser emitting at 488 nm. The capillary and nozzle of the flow cell had an inner diameter of 70 μm. The FSC signal, side scatter (SSC) signal, and MFI were simultaneously measured. Fluorescence emission was measured with a PerCP-Cy5-5 (FL2) channel, which had an emission wavelength near 710 nm. Phosphate-buffered saline was used as a sheath fluid, and the drop delay was calculated every time using fluorescent beads.

### 3.3. Fucoxanthin Assay

Pigments were extracted with ethanol and checked using silica plates (Merck TLC Silica gel 60, Darmstadt, Germany), using hexane/acetone (6:4) as the mobile phase [40]. Fucoxanthin content was measured with a Hitachi Primaide HPLC system (Hitachi, Tokyo, Japan) with a C18 reverse-phase column (2.7 μm particle size, 100 × 4.6 mm). The mobile phase consisted of acetonitrile and water with a flow rate of 1 mL·min^−1^. In the gradient condition, the acetonitrile/water ratio was increased from 80:20 to 100:0 in 8 min and maintained for 3 min and then decreased back to 80:20 in 5 min. The chromatogram was recorded at 445 nm. A fucoxanthin standard (ChromaDex, Irvine, CA, USA) was used for the construction of a standard curve in the 0.01–1.0 mg∙mL^−1^ range.

### 3.4. Screening Mutant Strains with High Fucoxanthin Content

Scatter plots were used to detect the fluorescence with the PerCP-Cy5-5 channel as the X-axis and SSC as the Y-axis. To verify the efficiency of the sorting setup, we used wild-type and transgenic strains with a resistance screening marker (*ble* gene) to perform flow sorting experiments. First, the wild type was treated with 50 μM DPA; the transformed strain was treated without DPA and so should have had a higher fucoxanthin content. A mixture of two strains was then used for flow sorting experiments. If our sorting setup was successful, the cells with high fluorescence would be those with zeocin resistance, which could form colonies on zeocin-containing plates (75 μM). Contrarily, cells with low fluorescence would be wild type and would not grow on the f/2 zeocin plates. Therefore, the sorted cells were counted and spread on f/2 plates containing 75 μM of zeocin, with 1000 cells per plate; every group used 5 plates. After 3 weeks, we counted the colonies grown on the plates and calculated the average.

For the mutant cell population obtained by physical mutagenesis, we used flow cytometry for high-throughput screening. We selected cells with the highest and lowest fluorescence values and sorted them with flow cytometry. The populations of these two gates were about 8% of the total cells; 1.5 × 10^6^ cells were sorted. After sorting, the cells were centrifuged to remove the sheath fluid and resuspended in 50 mL of fresh f/2 medium. The cells were cultivated under a photon flux density of 80 μmol photons·m^−2^·s^−1^. After 20 days of cultivation, the fucoxanthin contents of the mutant pools and wild-type cells were measured by HPLC.

### 3.5. Statistical Analysis

All the experiments were repeated three times. Unless otherwise stated, all data are expressed as mean ± standard deviation. The statistical significance and determination coefficients of the values obtained from each experiment were evaluated via multiple *t*-tests using the software GraphPad Prism 8.0.2. Significant differences were considered when *p* < 0.05.

## 4. Conclusions

We detected the spectrum peak of fucoxanthin at 710 nm in *P. tricornutum* using the excitation wavelength of 488 nm and found a linear relationship between fucoxanthin content and MFI. This method provides an efficient way to screen mutagenized cells and is a high-throughput approach for rapid analysis and real-time monitoring of cellular pigment content. Furthermore, the *P. tricornutum* strains with high fucoxanthin content were screened by flow cytometric sorting; the fucoxanthin content was 25.5% higher in mutants than in the wild type. This flow cytometric sorting technology provides an efficient and rapid method for screening cells highly enriched in fucoxanthin.

## Figures and Tables

**Figure 1 marinedrugs-19-00228-f001:**
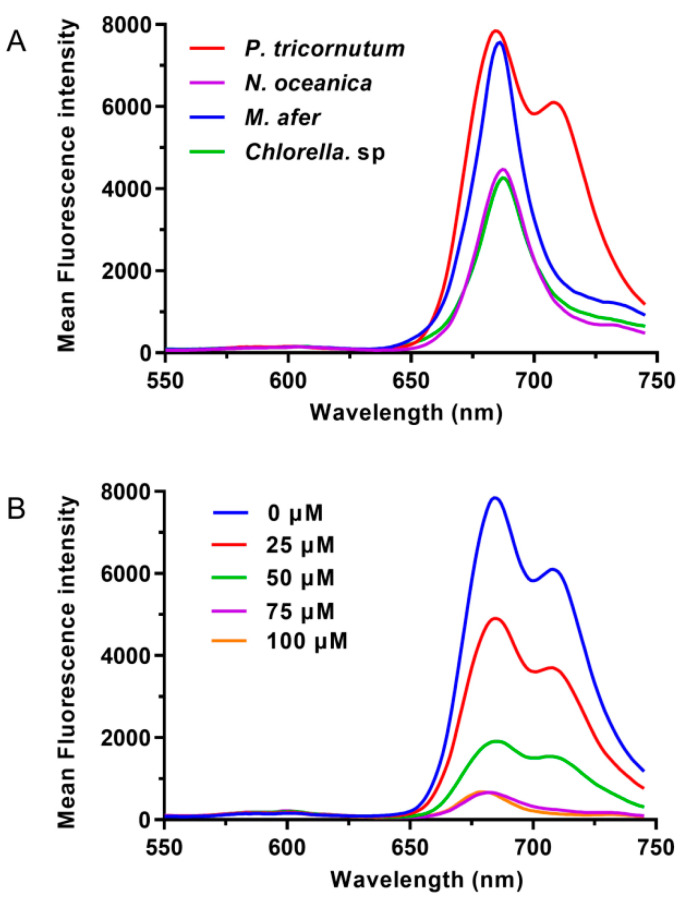
Fluorescence emission spectra of different microalgae strains (**A**) and *P. tricornutum* cultured with different concentrations of diphenylamine for 5 days (**B**). All the samples used the same OD_750_ value; the excitation wavelength is 488 nm.

**Figure 2 marinedrugs-19-00228-f002:**
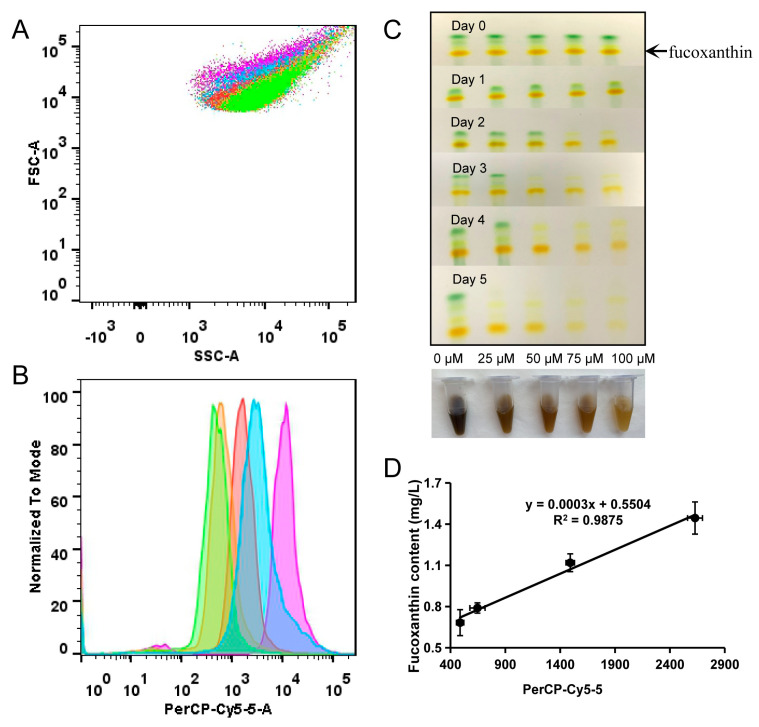
The fluorescence intensity and content of fucoxanthin in *P. tricornutum* cells treated with different concentrations of diphenylamine. (**A**) Scatter plot of the cells treated with different concentrations of diphenylamine by flow cytometry (SSC vs. FSC). (**B**) Histogram of the cells treated with different concentrations of diphenylamine (SSC vs. PerCP-Cy-5-5). Purple, blue, red, orange, and green points/lines in (**A**,**B**) represent cells incubated with 0, 25, 50, 75, and 100 µM diphenylamine, respectively. (**C**) TLC of the cells treated with different concentrations of diphenylamine in the culture period. (**D**) Correlation of fucoxanthin content and mean fluorescence intensity (MFI) in *P. tricornutum* cells. Each dot represents the cells treated by different concentrations of diphenylamine (25, 50, 75, and 100 µM). The fucoxanthin contents were determined using HPLC.

**Figure 3 marinedrugs-19-00228-f003:**
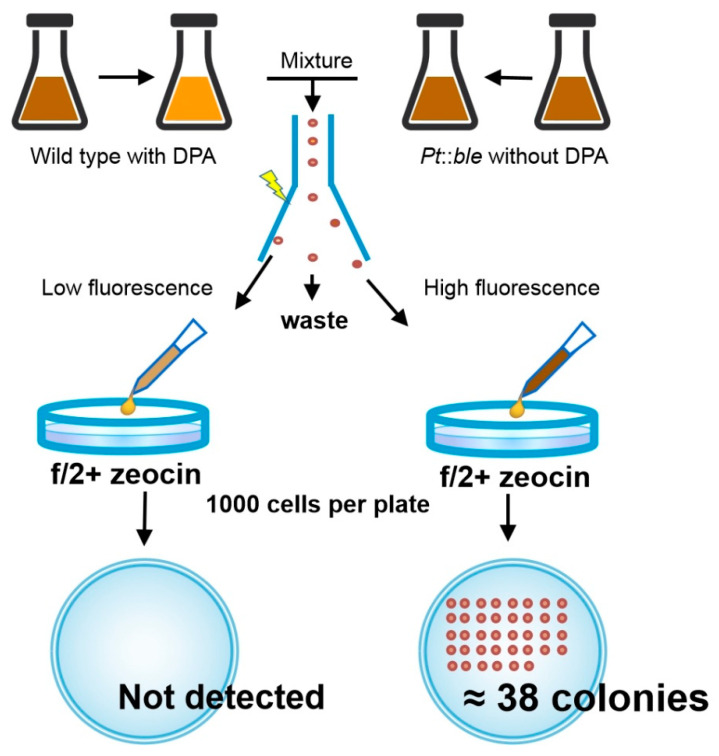
Schematic diagram of flow sorting of *P. tricornutum* wild type and mutant strains with different fucoxanthin content.

**Figure 4 marinedrugs-19-00228-f004:**
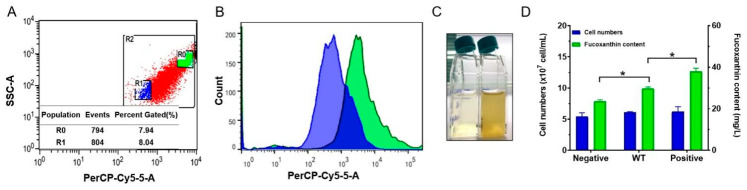
Cells were selected and sorted by flow cytometry. (**A**) R0 gate selected the cells with higher fluorescence (8%). R1 gate selected the cells with lower fluorescence (8%). R2 represents all the events. (**B**) The histogram shows the different mean fluorescence intensity (MFI) at the channel of PerCP-Cy5-5. (**C**) The negative mutations (left) and positive mutations (right) of *P. tricornutum* cells after 2 weeks’ culture. (**D**) Fucoxanthin content in the different samples. The positive group shows higher fucoxanthin content than the wild-type and negative groups. Error bars represent standard deviations, *n* = 3 (* *p*-value < 0.05).

## Data Availability

The data presented in this study are fully available in the main text and Appendix A of this article.

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
