# Peer review of "Rapid Sorting of Fucoxanthin-Producing Phaeodactylum tricornutum Mutants by Flow Cytometry"

_marinedrugs, 2021, doi:10.3390/md19040228_

Round 1

Reviewer 1 Report

the paper is ok for publication

Author Response

Thank you very much!

Reviewer 2 Report

The authors successfully amended the minor issues with the their manuscript.

They also implicitly suggetsed that the fluorescence emission peak is not related with fucoxanthim molecules themselves. Obviously, the emission originates from a long-wave Chl-fucoxanthin pigment-protein complex where the pigments are energetically coupled. At the same time, the paper lacks discussion on this topic which would be useful for the readership to better understand the underlying phenomena employed by the authors to monitor fucoxanthin content in the microalgal cells.

Author Response

Thank you for your suggestion.

We have been thinking about the problem you mentioned, and we are constantly checking the references to explain it. In addition to the previous responses, we have added new views from the references as support. And added the previous supplementary material data and references into the text. please check in line 65-93. 

Reviewer 3 Report

I reanalyzed the document and I consider that the authors improved the manuscript. However, some imprecisions in written English remains and the authors should improve it.

Author Response

Thank you for your suggestion.

 We reviewed the article carefully and hired a professional organization to revise the language of the article. I hope that the current version can fully explain our research. 

Round 2

Reviewer 2 Report

I believe that this manuscript is ready for pulishing now.

Author Response

Thank you very much.

This manuscript is a resubmission of an earlier submission. The following is a list of the peer review reports and author responses from that submission.

Round 1

Reviewer 1 Report

The authors describe the use of Phaeodactylum tricornutum, a diatom able to synthesise fucoxanthin, to develop a screening method to distinguish and separate organism with high and low content in fucoxanthin. They identified a secondary and unique spectrum peak at 710 nm and found a linear correlation between fucoxanthin content and the mean fluorescence intensity. Then they irradiated the P. tricornutum with heavy ions to create mutant organisms with high and low fucoxanhin content, then used flow cytometry to sort P. tricornutum mutants with different fucoxanthin contents. The analysis the authors developed is an effective, high-throughput approach to estimate fucoxanthin content in P- tricornutum. They propose an efficient, rapid,  and high-throughput approach to screen fucoxanthin-overproducing mutants.

The paper is well written and clear organised, the introduction gives all the necessary information to the reader to go through the different experiments performed, the results are linear, concise and exhaustive, the discussion leads to the clearly drawn conclusions of the study. The use of flow cytometry at the secondary spectrum peak for high-throughput screening and consequent selection of organism according to their level of fluorescent is very interesting and useful to isolate organism for following experiment.

Only one comment for the authors:

Line 128: an average of 38 colonies grew per plat: could you give more precise numbers or a standard deviation to understand the trend of the numbers.

Reviewer 2 Report

General comments

The paper by Fan et al. is dedicated to the development of a new cell sorting method for selection of the diatom cells rich in fucoxanthin. In principle, such a method would be welcomed by the specialists in the field of microalgal biotechnology. It seems that the emission band at 710 nm can be related to the increased level of fucoxanthin in the cells. However, I cannot accept that the fluorescence emitted around 710 nm really belongs to carotenoids and, in particular, to fucoxanthin. The quantum yield of carotenoid fluorescence is low, and its maximum is situated at shorter wavelength. It is much more probable that this emission peak originates from some long-wavelength form of chlorophyll. Anyway, the authors should check the emission spectrum of fucoxanthin before making strong a statement. Furthermore, it would not be surprising that intensity of F710 correlates with fucoxanthin content since the amount of Chl in the microalgal cells is in a stoichiometric relationship with that of primary carotenoids including fucoxanthin.

Specific comments

In abstract, the authors state that it the brown algae have a low fucoxanthin content but in the introduction, they say that the fucoxanthin is high.

L31: what is “clear genome”?

L34-35: it is not clear how “fucoxanthin synthesis” can “promote photosynthesis”.

LL67-68: what is the connection here between Chl and cytochrome synthesis?

L118: what is “early genetic transformation”?

L123: cells without the DPA inhibitor—not clear: cells containing no inhibitor or what?..

LL129-131: in this case, the genetically transformed nature of the high-fluorescence mutants should be also confirmed with the PCR of the plasmid (?) harboring the gene for zeocin resistance.

LL186-187: please provide the reference for identification of the Chlorella or include full details including primers, accession etc.

Reviewer 3 Report

In this manuscript, by Fan et al, reports a new method to evaluate fucoxanthin levels in producing mutants by fow cytometry. This carotenoid is of high relevance for the pharmaceutical industry and human health and this method can allow a fast screen of their levels in producing mutants, being potentially useful in HTS.

I consider that, despite the simplicity of this work, it can have high practical interest and is within the scope of Marine Drugs journal. However, in my opinion, some problems should be solved before being more acceptable for publication:

-the sentence in lines 31-33 should be improved for a higher clarity

-in the introduction, the authors should explain why heavy ion irradiation can lead to a higher production of fucoxanthin

-in lines 58-60 the authors introduce the use of DPA in the inhibition of phytoene conversion to β-carotene. This clearly needs adequate references

-in section 4.5 the authors developed the staristical analysis performed; however, in the text and figures, there is almost no use and evidence of these studies, namely t-tests

Reviewer 4 Report

I found this paper very interesting, with well-proven results. My remarks are:

  1. Did authors analyzed on LC other carotenoids in analyses?
  2. Authors wrote: "DPA was used to inhibit phytoene to β-carotene conversion...". Was it visible on performed  LC or TLC ?
  3. Was calculated  RF for TLC analyses for each carotenoids;
  4. Could authors point out zeocin on TLC plates?